# Ten Years of VIIRS On-Orbit Geolocation Calibration and Performance

Guoqing Lin [1,*], Robert E. Wolfe [1], Ping Zhang [2], John J. Dellomo [3] and Bin Tan [2]

1    Sciences and Exploration Directorate, NASA/GSFC, Greenbelt, MD 20771, USA
2    Science Systems and Applications, Inc., 10210 Greenbelt Road, Lanham, MD 20706, USA
3    Global Science & Technology, Inc., 7501 Greenway Center Drive, Suite 1100, Greenbelt, MD 20770, USA
*    Correspondence: guoqing.lin-1@nasa.gov; Tel.: +1-301-614-5451; Fax: +1-301-614-5269

**Abstract:** The first innovative Visible Infrared Imaging Radiometer Suite (VIIRS) sensor aboard the Suomi National Polar-orbiting Partnership (SNPP) satellite has been in operation for 10 years since its launch on 28 October 2011. The second VIIRS sensor aboard the first Join Polar Satellite System (JPSS-1) satellite has been in operation for 4 years since its launch on 18 November 2017, which became NOAA-20. Well-geolocated and radiometrically calibrated Level-1 sensor data records (SDRs) from VIIRS are crucial to numerical weather prediction (NWP) and Level-2+ environmental data record (EDR) algorithms and products. The high quality of Level-2+ EDRs is a requirement for the continuity of NASA Earth science data records (ESDRs) and climate data records (CDRs), one of the two objectives of the SNPP mission and one of the three elements in the JPSS mission objective. The other objective of the SNPP mission is risk reduction for the follow-on JPSS missions. This paper summarizes the on-orbit geolocation calibration and validation (Cal/Val) activities for both VIIRS sensors onboard SNPP and NOAA-20 in the past 10 years. These activities include nominal geolocation Cal/Val activities, risk reduction activities, and improvements for the on-orbit VIIRS sensor operations. After these activities, sub-pixel geolocation accuracy is achieved. Nadir equivalent geolocation uncertainty is generally within 75 m (1-σ), or 20% imagery band pixels, in either the along-scan or along-track direction for both SNPP and NOAA-20 VIIRS sensors. The worst 16-day measured geolocation errors (radial, 3-σ) are 280 m and 267 m, respectively, in the latest SNPP and NOAA-20 VIIRS data collections, which are better than the required accuracy of 375 m (radial, 3-σ). The risk reduction activities also improved VIIRS builds for JPSS-3 and JPSS-4 satellites, and provide lessons learned for other VIIRS-like sensor builds.

**Keywords:** VIIRS; SNPP; NOAA-20; ephemeris; attitude; pointing; geolocation; control point matching; error detection and correction; risk reduction

## 1. Introduction

The Visible Infrared Imaging Radiometer Suite (VIIRS) sensor was developed by inheriting features from several sensors that were in operation. The heritage includes Advanced Very-High Resolution Radiometer (AVHRR) on NOAA's Polar-Orbiting Environmental Satellites (POESs), Moderate-resolution Imaging Spectroradiometer (MODIS) on NASA's Earth Observing System (EOS) satellites, Sea-viewing Wide Field-of-view Sensor (SeaWiFS) on GeoEye's SeaStar satellite, and Operational Linescan System (OLS) on Defense Meteorological Satellite Program (DMSP) satellites [1–3]. In addition, new features were developed to limit pixel growth off-nadir [4].

This innovative VIIRS sensor was originally developed for the National Polar-Orbiting Operational Environmental Satellite System (NPOESS) [1,4]. The first VIIRS sensor was designated as part of risk reduction in the NPOESS Preparatory Project (NPP) with the local time at the ascending node (LTAN) at 13:30 [5,6]. NPP also provides data continuity for NASA Earth science data records (ESDRs) and climate data records (CDRs) derived from NASA's Earth Observation System (EOS) satellites (Terra and Aqua).

NPP was successfully launched on 28 October 2011. It was renamed Suomi National Polar-orbiting Partnership (SNPP) in early 2012 [7]. The NPOESS program was transitioned in 2010 to the Joint Polar Satellite System (JPSS) and the first JPSS satellite (JPSS-1, J01 or J1) was launched on 18 November 2017, subsequently renamed to NOAA-20 (or N20), carrying the second VIIRS sensor. Its orbit is in the same plane as the SNPP in the Earth center inertial (ECI) frame of reference, and has the same LTAN at 13:30 with a phase difference of a half orbit [3]. The SNPP and NOAA-20 altitude varies from 828 km near 15°N to 839 km near the North Pole and 856 km near the South Pole. A third VIIRS sensor onboard the JPSS-2 satellite will be launched in late 2022. The 4th and 5th VIIRS sensors onboard JPSS-3 and JPSS-4 satellites will be launched 5 and 10 years later, respectively.

The VIIRS sensor is a scanning imager that collects radiance from target scenes through a rotating telescope assembly (RTA) in the cross-track direction. A half-angle mirror (HAM) de-rotates the incoming rays from the RTA into a fixed aft-optics assembly (AOA). The AOA houses 5 imagery resolution bands (I-bands), 16 moderate resolution bands (M-bands), and a panchromatic day–night band (DNB). These bands cover a spectral range from 0.412 μm to 12.01 μm.

The nominal resolutions represented by horizontal sampling intervals (HSIs) at nadir are 375 m, 750 m, and 750 m for I-bands, M-bands, and DNB, respectively. There are 32, 16, and 16 detectors constituting each scan of I-band, M-band, and DNB, respectively. The I-bands are co-registered with M-bands by a 2 × 2 nesting scheme [8–10]. The growth of their pixel sizes on the ground is limited by aggregating three samples around nadir, two samples in the middle, and non-aggregation near edges of the scan in the scan direction (Figure 1). DNB aggregates charge-coupled device (CCD) sub-pixels in both scan and track directions in 32 zones, keeping the image pixel sizes nearly constant.

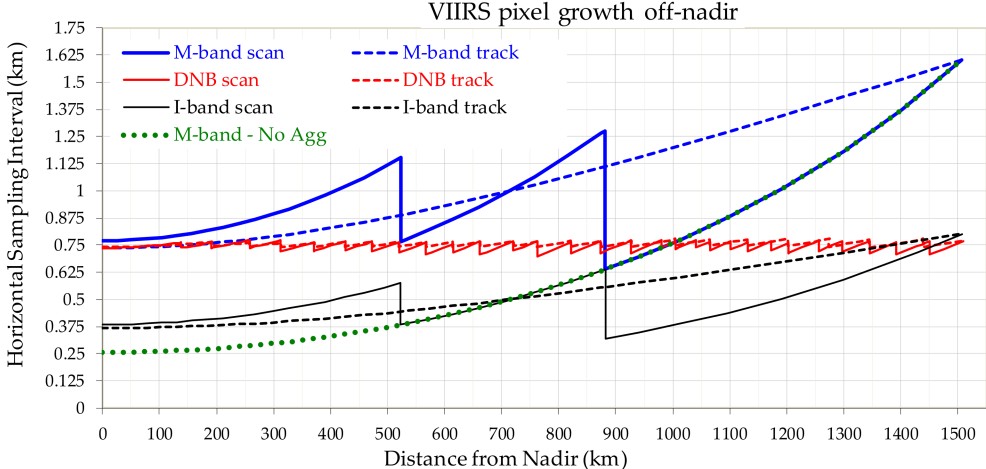

**Figure 1.** Dependence of pixel size on the off-nadir distance on the ellipsoidal Earth surface for a nominal build of VIIRS sensor. The curve for the "M-band—No Agg" in the plot is for dual gain M-bands in the scan direction before they are aggregated on the ground. Pixel growth in the track direction is unaffected by aggregation scheme for I-bands and M-bands. The pixel sizes are calculated using nominal instrument design values and satellite altitude of 828 km near the Equator (for improvement over MODIS pixel growth, see Figure 4 in [11]).

Well-geolocated VIIRS observations are crucial to NASA ESDRs and CDRs, and NOAA numerical weather prediction (NWP) and Level-2+ environmental data record (EDR) algorithms and products. The geolocation accuracy requirement states that known pointing of the VIIRS instrument boresight, following on-orbit calibration, shall be within 375 m (radial, 3-σ) of the true location of that field of view, at nadir, at any time during nominal operations. The on-orbit calibration corrects geolocation expressed in geodetic latitude and longitude coordinate point on the Earth surface and its associated terrain height for every sensor observation.

This paper provides an overview of the on-orbit geolocation calibration and improvement activities for VIIRS operations onboard SNPP and NOAA-20 satellites in the past 10 years. These activities corrected geolocation in long-term and annual variations, and resolved anomalies, and have achieved and surpassed the geolocation accuracy requirement. These activities also resulted in lessons learnt for the follow-on VIIRS sensors in the JPSS Program and other VIIRS-like sensors. Section 2 describes major methodologies. Section 3 describes nominal geolocation calibration processes immediately after launch and long-term monitoring and performance. Section 4 describes improvements and risk reduction procedures used to achieve the current performance. Discussion and future work are described in Section 5. Conclusions are detailed in Section 6. Many useful abbreviations used in this paper are listed in Appendix A.

## 2. Methods

A key component of geolocation calibration is a tool to detect geolocation errors. The tool is a control point matching (CPM) program. It uses a library of globally distributed ground control point (GCP) chips as ground truth. These chips were generated from high geolocation accuracy Landsat red band 30 m resolution images. They are heterogeneous, cloud-free, and high-contrast Landsat sub-images covering an area of about 20 km × 20 km. They are used to simulate coarser VIIRS band I1 images. The simulated images are correlated with band I1 images. By shifting the simulated images, a field of correlation coefficient values (CCVs) is obtained. The shifts at the maximal CCVs form geolocation residuals. A geolocation control point residual file is generated for every I-band geolocation granule. The mean values of the residuals constitute biases that are then used in look-up tables (LUTs) in a ground processing system to correct for geolocation errors, see details in [11,12].

The CPM program was originally developed for MODIS [13,14] and adapted for SNPP VIIRS [11,12,15] on-orbit geolocation calibration. The CPM uses a GCP library that has over 1200 image chips acquired from 1987 to 2003 by Landsat-5 and Landsat-7. Handling of the boundaries between the aggregation zones in the scan direction and between the scans in the track direction in the bow-tie deletion zones was improved [16]. The improved CPM was used immediately after NOAA-20 launch and in SNPP VIIRS re-processing. The CPM was further improved by refreshing the control point library using Landsat-8 images [17], and has resulted in improved accuracy [17–20].

The CPM program was executed in near real time as a product generation executable (PGE) at the NASA SNPP VIIRS Land Product Evaluation and Testing Element (PEATE), which was part of the risk reduction in the SNPP mission [21,22]. The Land PEATE transitioned to a Land Science Investigator-led Processing System (SIPS) after 2014. The team working on the Land PEATE or Land SIPS has been highly flexible and supportive of geolocation calibration activities. Every VIIRS data archive set comes with geolocation control point residual products. In addition, tests of the CPM improvements and geolocation performance improvements through LUT updates for VIIRS are carried out by the NASA Land PEATE or Land SIPS.

A copy of the CPM program is also run at the JPSS Government Resources for Algorithm Verification Independent Test and Evaluation (GRAVITE) and by colleagues at the NOAA Center for Satellite Applications and Research (STAR) at their facility [23–25].

The CPM detects band I1 geolocation errors, which are subsequently corrected through LUTs. Other I-bands and M-bands are aligned with band I1 [8–10].

For DNB, its samples are not co-registered with M-bands or I-bands. They are only roughly co-registered at nadir for the nadir samples. We have developed a method similar to the CPM program to detect VIIRS DNB geolocation errors by carefully selecting high-contrast Landsat-8 scenes [26]. The detected geolocation bias is corrected through changes in LUTs. This is usually carried out during the commissioning phase.

In addition to the automated CPM program, on-orbit operations are closely monitored for possible impacts on geolocation performance. The operations monitored include drag make-up maneuvers (DMUs), inclination adjust maneuvers (IAMs), leap-second insertions,

star tracker re-alignment, onboard clock drift corrections, and VIIRS instrument lunar roll calibration maneuvers. Some impacts from these operations are described in Section 4.

### 3. On-Orbit Geolocation Calibration and Validation

Before launch, the VIIRS instruments went through intensive ground testing. Geometric testing included sensor spatial response, band-to-band registration, and alignments and pointing tests [8–10]. Datasets from alignment and pointing tests are used in ground processing software [11,27,28] for on-orbit geolocation data product generation. Some parameters from these tests are used in LUTs, which were prepared before launch. The "at-launch" geolocation parameter LUTs are shown in Table 1 that were prepared on 12 December 2011 for SNPP VIIRS, and in Table 2 that were prepared on 18 November 2017 for NOAA-20 VIIRS. Some geolocation parameters are tuned in on-orbit operations for accurate geolocation performance [11,13,16,29]. This section describes the initial on-orbit geolocation calibration and the current geolocation performance. Section 4 describes the major improvements, which result in better geolocation performance as we describe in Section 3.2.

**Table 1.** List of major SNPP VIIRS geolocation parameter updates.

| Parameter (arcsec) | At-Launch 12 December 2011 | 1st Update 23 February 2012 | 2nd Update 8 April 2013 | 3rd Update 3 August 2013 | VIGMU [1] + RPY(t) [2] 13 November 2020, C2 |
|---|---|---|---|---|---|
| inst_roll | 33.2 | −227.3 | −227.3 | −222.8 | −107.2 |
| inst_pitch | 41.2 | 153.2 | 66.4 | 59.1 | 60.5 |
| inst_yaw | −59.3 | 95.4 | 74.4 | 78.4 | 51.4 |
| ham_alpha | 3.9 | 3.9 | 3.6 | 3.6 | −2.8 |
| ham_beta | 9.5 | 9.5 | 9.5 | 9.5 | −12.3 |
| ham_gamma | −6.0 | −6.0 | −6.2 | −6.2 | 6.3 |
| ham_roll | 0.0 | 0.0 | 21.8 | 21.8 | −14.6 |
| ham_pitch | 0.0 | 0.0 | 79.7 | 79.7 | 35.2 |
| ham_yaw | 0.0 | 0.0 | 38.8 | 38.8 | −124.2 |
| tele_roll | 0.0 | 0.0 | 0.0 | 0.0 | 0.0 |
| tele_pitch | 0.0 | 0.0 | 65.9 | 65.9 | 65.9 |
| tele_yaw | 0.0 | 0.0 | 11.0 | 11.0 | 11.0 |

Notes: NASA Land SIPS started re-processing Collection-2 Level-1 data products on 13 November 2020 and released publicly on 24 August 2021. Additional notes are described below. [1] VIGMU = VIIRS instrument geometric model update. [2] RPY(t) represents additional time series of correction in the inst2sc roll, pitch, and yaw (the first three rows in the table).

**Table 2.** List of major NOAA-20 VIIRS geolocation parameter updates.

| Parameter (arcsec) | At-Launch 18 November 2017 | 1st Update 3 January 2018 | 2nd Update 14 March 2018 | VIGMU + RPY(t) 12 July 2019, C2 | VIGMU + RPY(t) 20 January 2021, C2.1 |
|---|---|---|---|---|---|
| inst_roll | 0.9 | −423.5 | −422.9 | −363.6 | −363.6 |
| inst_pitch | 51.1 | 300.5 | 298.5 | 295.4 | 295.4 |
| inst_yaw | 80.5 | 99.4 | 111.4 | 112.6 | 112.6 |
| ham_alpha | 3.9 | 3.9 | −4.4 | 9.2 | 9.2 |
| ham_beta | 9.5 | 9.5 | 9.5 | −15.9 | −15.9 |
| ham_gamma | −6.0 | −6.0 | 1.0 | −2.6 | −2.6 |
| DNB FPA x-offset (m) | 0.00000 | −0.00380 | −0.00403 | −0.00403 | −0.00403 |

Notes: NASA Land SIPS started re-processing Collection-2.1 Level-1 data products on 20 February 2021 and publicly released them on 24 August 2021.

### 3.1. VIIRS Initial On-Orbit Geolocation Calibration

After the SNPP was launched on 28 October 2011, the spacecraft and instruments went through early orbit checkout (EOC) for a few weeks. The last orbit raising maneuver was performed on 16 November 2011. The onboard attitude determination and control system (ADCS) was calibrated and tuned on 17 November 2011 [13]. The VIIRS nadir door was opened on 21 November 2011, making VIIRS data from the visible and near-infrared bands available. Due to a discovery and investigation of VIIRS RTA degradation [30], which took about two months, the cryo-radiator door was opened on 18 January 2012 to cool down the focal plane assemblies (FPAs), making data from all VIIRS bands available.

For NOAA-20, after its launch on 18 November 2017, it was inserted in the same orbital plane as the SNPP but 8 km lower and behind SNPP. It caught up and flew ahead of SNPP because it was 4 m/s faster initially [16]. Several orbital raising maneuvers were performed to achieve the same altitude as SNPP, but half of an orbit, or 50.75 min, ahead of SNPP. The last orbit raise was performed on 6 January 2018 with small adjustments in altitude and velocity. The ADCS was calibrated on 20 November 2017. However, the time tag for attitude (quaternion) is 0.1 s behind the time tag for the ephemeris. The attitude determination and control mistakenly used ephemeris data 0.1 s ahead in the flight software which caused a pitch offset of 21.3 arcseconds. A LUT in the flight software was adjusted on 7 February 2018 to change the frame of reference of the spacecraft coordinate system (relative to the geodetic orbital coordinate system), which resulted in near-zero pitch values as computed by the ground software. Such an adjustment does not change nadir pointing and geolocation for instrument data products [16].

The initial geolocation for VIIRS on SNPP and N20 has large errors in both scan and track directions (Figure 2). They were up to about 1000 m for SNPP VIIRS and 2000 m for N20 VIIRS. These errors are expected due to large tolerance in instrument-to-spacecraft mounting alignment before launch and shifts during launch.

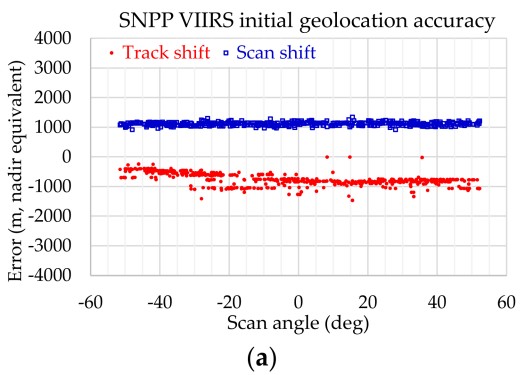 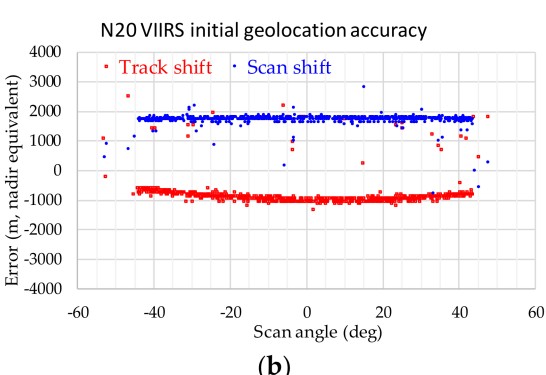

(**a**)　　　　　　　　　　　　　　　　　　　　　(**b**)

**Figure 2.** Initial geolocation accuracy for (**a**) SNPP and (**b**) N20 VIIRS sensors.

The error detection CPM program was configured with step size of 0.2 I-band sampling intervals (~75 m at nadir) at the beginning of the on-orbit operations, in anticipation of possible large geolocation errors right after launch. The CPM program uses a search area of ±50 steps in both scan and track directions. That enables the CPM to measure geolocation errors up to 3750 m nadir equivalent in the positive and negative scan and track directions. After the initial geolocation errors are corrected for, the step size was configured back to nominal 0.05 I-band sampling intervals (~19 m at nadir) for long-term monitoring.

### 3.2. Long-Term Monitoring and Improvements in Geolocation Performance

The initial on-orbit geolocation errors were corrected through instrument-to-spacecraft (inst2sc) interface rotation angles in LUTs. These angles are Euler roll, pitch, and yaw (RPY) angles, coinciding and having the same effect of spacecraft attitude angles measured by ADCS and downlinked to the ground for geolocation determination. These corrections

were applied on 23 February 2012 for SNPP VIIRS (Table 1) and on 3 January 2018 for N20 VIIRS (Table 2). Further improvements were applied, see Section 4.

Table 1 lists four (4) major geolocation parameter LUT updates. The first three updates were delivered to the Interface Data Processing Segment (IDPS) of the SNPP Program in forward processing. The latest update was delivered to NASA VIIRS Land SIPS in mission-wise re-processing and continuing forward processing. The parameters include instrument-to-spacecraft mounting interface ration angles (inst_roll, inst_pitch, inst_yaw), HAM wedge angles (ham_alpha, ham_beta) and HAM motor axis angle (ham_gamma), HAM alignment (ham_roll, ham_pitch, ham_yaw), and rotating telescope assembly (RTA) alignment angles (tele_roll, tele_pitch, tele_yaw).

An additional parameter is the DNB detector position on the focal plane (similar to "DNB FPA x-offset" in Table 2) that was adjusted by a Northrop Grumman (NG) team on 30 March 2012 to further remove 1.3 km nadir equivalent bias in the scan direction. The parameter was refined on 15 February 2013 by the same NG team to make the geolocation accuracy closer to the truth by about 340 m nadir equivalent in the scan direction [11,31].

As we can see from Table 1, the "1st Update" has large values in initial corrections. The "2nd Update" is refinement with additional internal orientation parameters that are used to correct for biases in scan angle profiles, see details in [11]. The "3rd Update" was in response to an event described in Section 4.2.2. The latest updates are described in Sections 4.3.2 and 4.3.4.

Table 2 lists four (4) major updates for the NOAA-20 VIIRS geolocation parameter LUTs. This time, we worked with the NOAA STAR colleague to deliver the updates to IDPS [24] in forward processing. We also worked out a method for an additional parameter updates for "DNB FPA x-offset" [26]. NASA Land SIPS re-processed the data from the start of mission upon receiving the updates.

Similar to Table 1 for SNPP VIIRS LUT updates, the "1st Update" NOAA-20 VIIRS LUT has large initial corrections. The "2nd Update" has refined values. The latest updates will be described below and in Sections 4.3.2 and 4.3.4.

The last columns in Tables 1 and 2 represent the LUTs updated after implementing the VIGMU. The updates also use refreshed GCP chips from Landsat-8 for the first time. "RPY(t)" represents additional time series of correction in the instrument to spacecraft interface mounting rotation angles (inst_roll, inst_pitch, and inst_yaw) (the first three rows in the table). The code in ground geolocation processing software to accept the additional temporal variation of pointing correction in the existing LUTs was first implemented in NOAA-20 VIIRS data Collection-2 (C2) on 12 July 2019 by the NASA Land SIPS [29], as indicated in the second to last column in Table 2. The RPY(t) for SNPP VIIRS C2 is shown in Figure 3. The data were generated based on geolocation errors detected from the previous data collection C1. In forward processing, though, the pointing correction values are constant, using the latest available bias values. We monitor the trend closely, and update as needed. The figure marks the following six major events:

- "A": Switch of VIIRS scan control electronics (SCE) from Side-B to Side-A on 22 November 2012;
- "B": Erroneous star tracker realignment on 25 April 2013;
- "C": 1 s time error onboard of SNPP (213 arcsec pitch error) for ~7 h on 19 August 2015;
- "D": Star Tracker-2 reset on 22 March 2019;
- "E", "F": Forward processing with constant pointing correction onward from 8 February 2021, 13 January 2022, respectively.

The additional time series of pointing correction RPY(t) for NOAA-20 VIIRS is shown in Figure 4. The data were generated based on geolocation errors detected from the previous data collection C2. As we can see, there are regular annual variations in roll and pitch lately. We extended the correction in forward processing for one more year after mid-2021. In mid-2022, that extended prediction of pointing correction was confirmed by the CPM in C2.1. Another one-year extension of the LUTs was put into operation in C2.1 (Figure 4) as well as in C2, which was based on geolocation errors detected from the latest data collection

C2.1 in the previous year. In contrast, the pointing variations in SNPP VIIRS do not have such patterns.

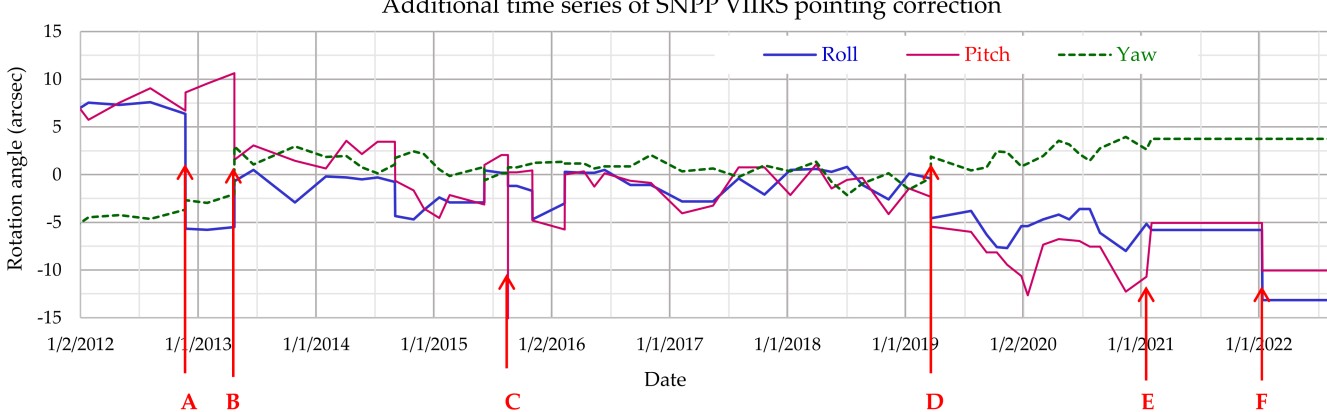

**Figure 3.** Temporal variations in pointing correction through additional instrument-to-spacecraft mounting interface rotation angles for SNPP VIIRS C2. The labels denote the known events that impacted geolocation. A: Side switch of VIIRS scan control electronics; B: Erroneous star tracker realignment; C: 1 s time error onboard of SNPP (213 arcsec pitch error) for 7 h; D: Star Tracker-2 reset; E and F: Forward processing with constant pointing correction.

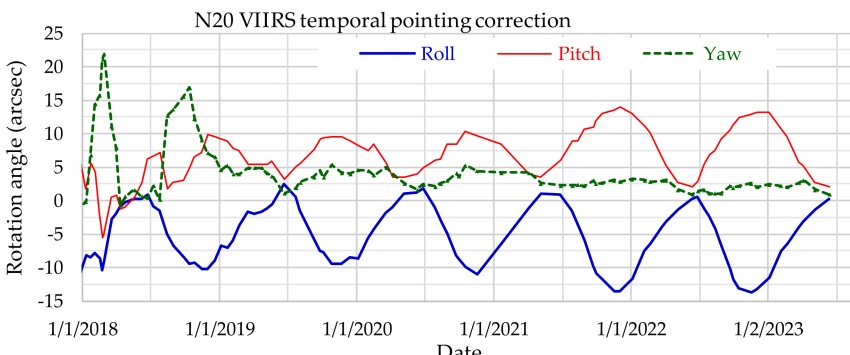

**Figure 4.** Temporal variations in pointing correction through additional instrument-to-spacecraft mounting interface rotation angles for NOAA-20 VIIRS C2.1.

With pointing corrections shown in Table 1 and Figure 3, the latest geolocation accuracy performance for SNPP VIIRS in NASA Land SIPS C2 is shown is Figure 5. Note that the shifts are detected using the refreshed Landsat-8 chip library, while the correction in Figure 3 was generated from the biases detected using the existing (old) chip library before February 2021. These two libraries are slightly different in performance [17]. The remaining small biases will be further corrected for in future data collections.

Notice that the latest event "H" in Figure 5 resulted from a timestamp error at the end of 27 November 2021, when the Global Positioning System (GPS) had issued a "0" leap second since it had been 256 weeks since the last leap second insertion at the end of 31 December 2016. This event is found to somehow impact VIIRS instrument pointing by about 5 arcseconds in both scan and track directions. We corrected the induced geolocation errors in the forward-processing data stream on 13 January 2022 (mark "F"). Such biases will be further corrected for in future data collections.

Figure 6a shows the latest geolocation accuracy performance for NOAA-20 VIIRS in NASA Land SIPS C2.1. Figure 6b shows what the performance would be without temporal pointing correction as shown in Figure 4. The geolocation performance in earlier collections can be found in earlier reports or papers [16,29]. The variation becomes more regular after initial fluctuation. The large variation prompted us to accelerate the implementation of temporal pointing correction in mid-2019 [29]. The regular annual variation enables us to

more accurately predict temporal pointing correction in forward processing as shown in Figure 4.

**Figure 5.** Measured SNPP VIIRS geolocation performance in Collection-2 in daily means, 16-day means, and 16-day uncertainties. "G" denotes an event on 3 August 2021 when the instrument was put in "safe hold" and the spacecraft was "sun pointing" for about 10 h that impacted thermal conditions of the instrument, which in turn impacted pointing and geolocation. "F" denotes LUT update on 13 January 2022 after an unexpected event of pointing shifts on 28 November 2021 ("H").

The overall geolocation performance for SNPP and NOAA-20 VIIRS sensors is tabulated in Table 3. In the most recent re-processing collections, SNPP C2 and NOAA-20 C2.1 started using the refreshed GCP library to monitor the geolocation errors. Their corresponding earlier collections, SNPP C1 and NOAA-20 C2, use the existing (old) GCP library. It is worth pointing out that the refreshed chip library generated over 4 times the matched points from the old library. NASA Land SIPS continues processing the earlier collections with a long enough overlap with the newest collections so that users can have enough time to switch to the newest collections of data products from NASA Land SIPS.

**Table 3.** Overall SNPP and NOAA-20 VIIRS geolocation accuracy in nadir equivalent units.

| Residuals | SNPP C1 | NOAA-20 C2 | SNPP C2 | NOAA-20 C2.1 |
|---|---|---|---|---|
| Track mean (m) | 13 | 0 | 2 | −4 |
| Scan mean (m) | 5 | 1 | 7 | 6 |
| Track RMSE (m) | 59 | 55 | 59 | 57 |
| Scan RMSE (m) | 52 | 49 | 48 | 47 |
| Data-days (years) | 3519 (9.6) | 1341 (3.7) | 3743 (10.2) | 1565 (4.3) |
| Missing days | 1 | 3 | 1 | 3 |
| Daily matches | 202 | 195 | 851 | 841 |

Notes: NASA Land SIPS keeps processing earlier collections of data products until all users have transitioned to using latest collections. So, the SNPP C1 and NOAA-20 C2 will co-exist with SNPP C2 and NOAA-20 C2.1 for a while.

Figures 5 and 6a show that in 16-day measurement statistics, nadir equivalent geolocation uncertainty is generally within 75 m, or 20% nominal imagery band pixel size of 375 m, in either the along-scan or along-track direction for both SNPP and NOAA-20 VIIRS sensors. The mission-wise performance in Table 3 shows smaller geolocation uncertainties in root mean square errors (RMSEs) (less than 60 m) due to overall averaging effect.

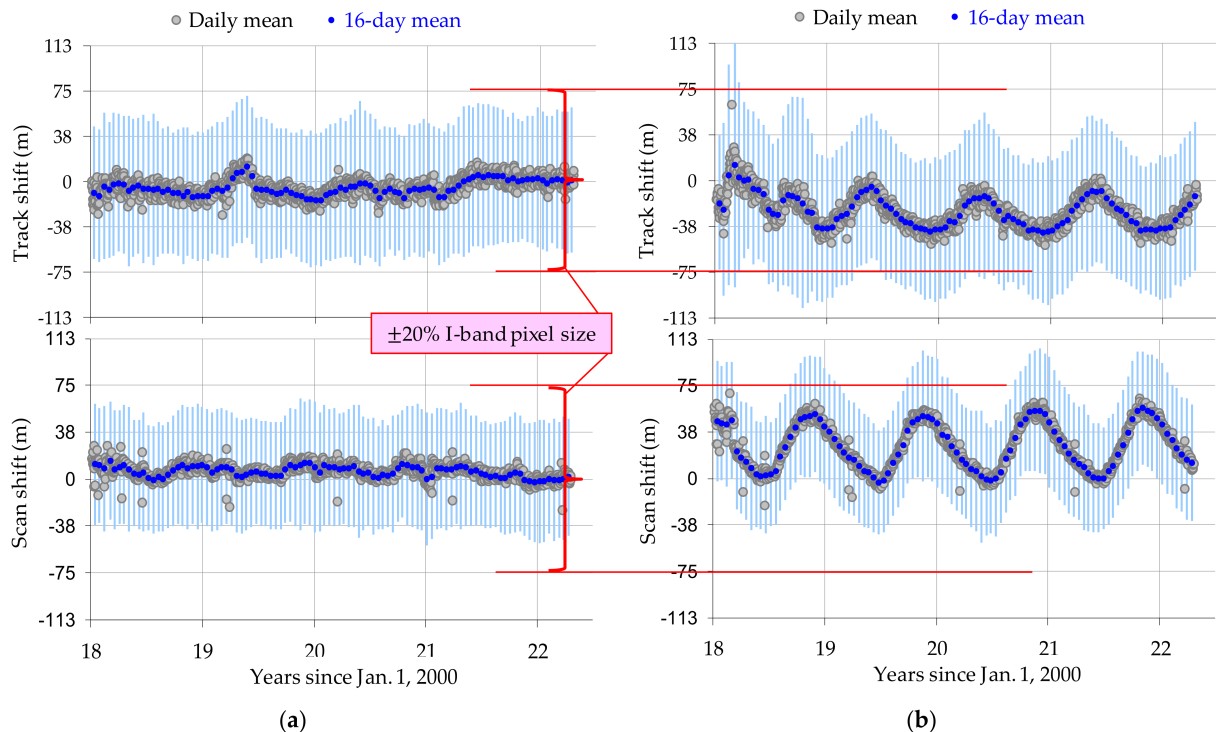

**Figure 6.** Measured NOAA-20 VIIRS geolocation performance in Collection-2.1 in daily means, 16-day means, and 16-day uncertainties. (**a**) Detected from the operation geolocation data products; (**b**) what the performance would be without temporal pointing correction.

To gauge the performance against the requirement of geolocating observations within 375 m (radial, 3-σ) of the true location at any time during nominal operations, we use the worst case "16-day" measurement statistics as a proxy of the "any time" probability. In our 16-day statistics, we have means and standard deviations in the scan and track directions. We take the root sum squared (RSS) of the means in the scan and track directions as the radial mean and RSS of the corresponding standard deviations as the radial standard deviation. Table 4 shows the worst case radial mean and radial standard deviation in the first two rows. The last row shows the worst case combination of three times the standard deviation on top of radial mean, which we take as a measure of geolocation uncertainty in the radial direction at the 3-σ level. We see that the measured radial means move closer to 0 as the data collection progresses from C1 to C2 for SNPP VIIRS, and from C2 to C2.1 for N20 VIIRS. The measured radial standard deviation also decreases as the data collection progresses, and the geolocation accuracy improves as the uncertainty becomes smaller, moving further within the 375 m requirement.

**Table 4.** Worst case 16-day statistics in geolocation error measurements.

| Residuals | SNPP C1 | NOAA-20 C2 | SNPP C2 | NOAA-20 C2.1 |
|---|---|---|---|---|
| Radial mean (m) | 53 | 27 | 36 | 20 |
| Radial stdev (m) | 114 | 87 | 91 | 86 |
| Radial accuracy (3-σ, m) | 363 | 279 | 280 | 267 |

For N20, if the correction for temporal pointing variations was not performed, the worst case radial mean, radial standard deviation, and 3-σ geolocation uncertainty would be 70 m, 111 m, and 383 m, respectively. That would not meet the 375 m requirement.

This performance in SNPP VIIRS C2 and NOAA-20 C2.1 is accomplished through improvements, and some risk reduction procedures described in Section 4.

## 4. On-Orbit VIIRS Geolocation Performance Improvements

The VIIRS sensor was built upon the heritage but a mixture of features from AVHRR, MODIS, SeaWiFS, and OLS [1–3]. There are also new features [4]. As such, it was expected that some unexpected improvements were needed, which was the rationale behind the designation of NPP as a risk reduction project for the JPSS Program. The on-orbit geolocation performance improvements have been attained through three major classifications of solutions. The first is simply to mask or hide the problematic data. This solution is the simplest to achieve in most cases, but is the least preferable as it is simply throwing away data. More details on this solution and some examples of its use are provided in Section 4.1. The second solution is to implement changes in the LUT parameters and is described in more detail in Section 4.2. These two solutions do not cover all improvements that have been performed; any remaining solutions are handled with modifications of the geolocation algorithms. This last, more general, case is described in more detail within Section 4.3.

### 4.1. Solution 1: Mask the Data

In certain situations, it is prudent to simply remove problematic data. The method of removal depends on the nature of the data and the problem and includes hiding of granules, setting geolocation values to be a fill value, and/or creation/setting of quality flags. This solution is used sparingly in order to retain as many data as is feasible.

#### 4.1.1. Loss of Synchronization from HAM to RTA

The VIIRS HAM is designed to rotate synchronically at half speed of the RTA motion. In on-orbit operations, however, HAM occasionally loses synchronization with the RTA. Fortunately, the scan control electronics (SCE) is designed to restore the synchronization by adjusting the rotation speed of the HAM. The duration of the restoration usually takes about 100 s for "long" synchronization (sync) losses. Occasionally, "short" sync losses are restored quickly in about 15 s.

SNPP VIIRS was initially operated on the SCE B-side. The first sync loss event was identified on 18 November 2011, 20 days after launch. The sync loss events occurred about once every 17 days. The SCE was switched to A-side on 22 November 2012. The frequency has been lower, at about once every 30 days on SCE A-side. There are 127 "long" and seven "short" sync loss events for the SNPP mission up to June 2022. NOAA-20 VIIRS had the first sync loss event on 1 December 2017, 13 days after launch. The frequency is about once every 20 days. There are 74 "long" and five "short" sync loss events for the N20 VIIRS mission up to June 2022.

When RTA/HAM synchronization loss occurs, pointing accuracy is lost. A mitigation method in the ground processing system was to mask the fields in the geolocation and radiometric data products, and a quality flag was added in scan attributes. That was accomplished in near-real time (NRT) in IDPS and in NASA Land PEATE in 2013.

#### 4.1.2. Sector Rotation for Lunar Calibration Maneuvers

The VIIRS instrument has an extended opening before the start of scan of the "Earth View" (EV), used as a "Space View" (SV) calibration port [32,33]. The SV was also originally designed to acquire the Moon during lunar roll calibration maneuvers. By design, the sensor collects samples covering an angular space of 0.86 degrees while the Moon has the size of 0.49 to 0.56 degrees and would be fully covered by these calibration samples if the center of the Moon is at the middle of the SV. However, these SV samples are collected without band co-registration, so these samples cover different angular spaces for different bands.

The SNPP conducted the first lunar roll calibration maneuver (LRM) on 4 January 2012 [34]. Samples for M6 covered the Moon while samples for M1 covered partial Moon, as mentioned above. An improved LRM was performed on 3 February 2012 in two consecutive orbits. That enabled the full coverage of the Moon by the SV samples from all bands, but the coverages of the Moon were skewed for some bands without enough samples in the

background which hindered full characterization of these bands [34]. Further improved mitigation was implemented in the LRM on 2 April 2012 when the sector of the EV was rotated. The middle of the EV samples was moved to the center of the SV that targeted the center of the Moon. All bands are co-registered in the EV, and thus the Moon data were also co-registered with this modified maneuver. However, single gain bands are aggregated by three native samples in the scan direction. In NOAA-20 VIIRS LRMs, the EV sector is rotated so that the middle of the non-aggregated EV samples targets the Moon, which is covered by native (unaggregated) samples for all bands.

The sector rotation is performed by changing the encoder at the start of scan for the RTA. As expected, the pointing has a large offset and geolocation of the EV samples is wrong. Subsequently, the ground processing system applied masks to the fields in the geolocation and radiometric data products, and a quality flag is added in scan attributes.

### 4.1.3. Trailing Effects after Active Orbit Management Operations and Delayed Leap-Second Insertions

The SNPP and NOAA-20 spacecrafts carry Global Positioning System (GPS) receivers and incorporate ephemeris data from GPS signals in the orbit propagators to ensure accurate spacecraft ephemeris data for proper instrument data geolocation. However, that incorporation process is slow. When an IAM occurs, there is a rapid change in velocity that takes the orbit propagator about two hours to absorb. For a DMU maneuver, the difference in along-track velocity is much smaller than the error from IAMs, but does still take the orbit propagator time to absorb. These errors are gradually reduced as the orbit propagator incorporates ephemeris from GPS signals, converging after about 2 h.

When a leap second was inserted at the end of 30 June 2012, 30 June 2015, and 31 December 2016, a position error of about 6.5 km on the ground was observed for SNPP when the time and corresponding ephemeris are not adjusted onboard the spacecraft in time. In the case of leap-second insertion at the end of 30 June 2012, timestamps in one-second intervals were repeated within the first one minute on 1 July 2012. However, the corresponding satellite ephemeris (position and velocity) were not treated properly so the position was about 7.5 km behind what it should be. That error was gradually reduced as the orbit propagator incorporated ephemeris from GPS signals. The convergence took about 2 h.

We worked with the NOAA Mission Operations Team (MOT) to implement ways in orbit operations to minimize these errors, or the durations of the errors, and simply hide the data in NASA Land SIPS for which the errors are unsolvable.

### *4.2. Solution 2: Revise the LUT*

In some cases, the algorithm remains correct but various modifications of the parameters stored in the LUT need to be modified. This solution covers a variety of cases depending on the parameters changed.

### 4.2.1. Switch of SNPP VIIRS SCE Side

As mentioned in Section 4.1.1, the scan control electronics (SCE) was switched from B-side to A-side on 22 November 2012 to reduce the frequency of RTA/HAM synchronization losses. However, the switch induced geolocation errors of ~325 m in the scan direction because the encoder at the start of scan for HAM is different in SCE A-side than in B-side. The geolocation parameter LUTs incorporated changes to the polynomial coefficients for telescope encoder-to-angle conversion from one to two dimensions. The updated LUTs went into operation in IDPS on 11 December 2012.

### 4.2.2. SNPP Star Tracker Re-Alignment

SNPP mission operations adjusted orientation on one of the two star trackers on 25 April 2013. The pointing vector of that star tracker was adjusted by about 12 arcseconds in roll, pitch, and yaw, resulting in a jump of about 6 arcseconds to the nadir point of the

spacecraft and its instruments. VIIRS CPM detected the jump and after a few rounds of communications and tests for changes in LUTs by the NASA Land PEATE, the updated LUTs went into operation in IDPS on 3 August 2013.

This event was taken as a lesson learnt. The mission operations do not make such unbalanced adjustments again to the star tracker vectors in nominal operations for both SNPP and NOAA-20 missions. No such star tracker re-alignment should be made for future JPSS missions either.

### 4.2.3. SNPP Spacecraft Control Clock Error

The SNPP onboard control clock incurred an error of 1 s on 19 August 2016 at 14:23 UTC. This resulted in a 1 s discrepancy in the spacecraft ephemeris data (position and velocity) and the ADCS, controlling effectively one second behind the ephemeris data. This is an interesting problem, similar to what happened to NOAA-20 initial attitude control when attitude control was 0.1 s ahead of ephemeris data that resulted in a 21.3 arcsecond offset in pitch [16]. The mission operation corrected for the time error at 21:12 UTC the same day. The time error caused a VIIRS geolocation error of about 850 m in the track direction during a period of 6 h 49 min. SNPP VIIRS C2 implemented the correction in the time series of pointing variation.

### 4.2.4. Focal Length Measurement with On-Orbit Data

Following the example of the CPM algorithm [11,14], we selectively picked Landsat-8 cloud-free, high-contrast scenes at times close to VIIRS scenes. We first correlated with VIIRS sub-images within a scan to Landsat sub-scenes by shifting Landsat scenes as measures of VIIRS geolocation errors. We then performed tests using multiple different focal length values as stored in LUTs. The focal length that yielded the best correlation was determined as the measured on-orbit focal length. The results show that, indeed, SNPP VIIRS has a shorter effective focal length (EFL) (at about 0.5%) while NOAA-20 VIIRS remains at its nominal EFL value, with uncertainties within 0.1% [26]. We then modify the LUTs to use the measured EFL for operational data product generation.

### *4.3. Solution 3: Improve the Algorithm*

Occasionally, corrections to the geolocation algorithm are required. These modifications tend to be larger in the scale of the change than the previous types of improvements, though the actual error correction is not always larger.

### 4.3.1. Terrain Correction for DNB Geolocation

At the time of SNPP launch, VIIRS geolocation data products included I-band, M-band, and DNB ellipsoidal geolocation products, as well as I-band and M-band terrain corrected geolocation products. The DNB did not include terrain corrected geolocation products [11,31], which have geolocation errors monotonically increasing with increasing off-nadir scan angle and terrain height (Figure 7). The geolocation error was more than 1 km for scan angles over 40° over every 1 km of terrain height. We implemented the code similar to those used in I-band and M-band terrain corrected geolocation in mid-2013 in NASA Land PEATE and on 22 May 2014 in IDPS [35,36]. The code changes in IDPS were applied to the forward-processing data stream. In NASA Land PEATE, and later in NASA Land SIPS, code changes were applied to forward-processing data streams as well as in re-processing for the whole mission. NOAA STAR has also applied the code changes in re-processing geolocation products [25].

When down-stream products are retrieved from the non-terrain corrected geolocation products, the geolocation errors are carried over. That happened to VIIRS imagery EDRs in which the mountains appeared to be moving in the overlapping areas in two adjacent passes [37]. Code changes for terrain corrected geolocation were applied to imagery EDRs on 28 July 2020 for both SNPP and NOAA-20 VIIRS sensors in the forward-processing data streams [38].

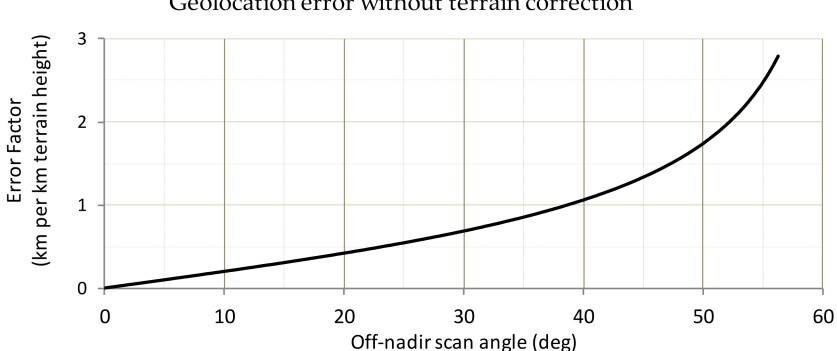

**Figure 7.** VIIRS parallax geolocation error from the ellipsoidal Earth model surface to the actual Earth terrain surface every one km of terrain height.

### 4.3.2. VIIRS Instrument Geometric Model Update (VIGMU)

The VIIRS geolocation errors have a strange pattern of fluctuation with respect to the scan angle out-of-phase (180 degrees) from the pattern of non-linearity of HAM motion. The ground geolocation software makes full use of encoders in the scan pointing knowledge equation, so this fluctuation pattern should not have appeared in the geolocation errors. Re-analysis of pre-launch test data and documents revealed that images entering the RTA flip before they are reflected by the HAM and enter the aft-optics assembly. This effectively changes the sign of magnification, currently +4.0, used in the ground software. We confirmed with the instrument vendor that the magnification should be −4.0. We made the changes and call this the VIIRS instrument geometric model update (VIGMU). The VIGMU has been implemented in the NASA Land SIPS for SNPP C2, NOAA-20 C2, and NOAA-20 C2.1. It has also been implemented in the IDPS in forward processing since 4 February 2021.

The results of this change are shown in Figure 8. Prior to incorporation, there is an oscillatory component to the errors in the scan direction, as a function of the observed scan angle. After the VIGMU was incorporated, this oscillatory component was eliminated.

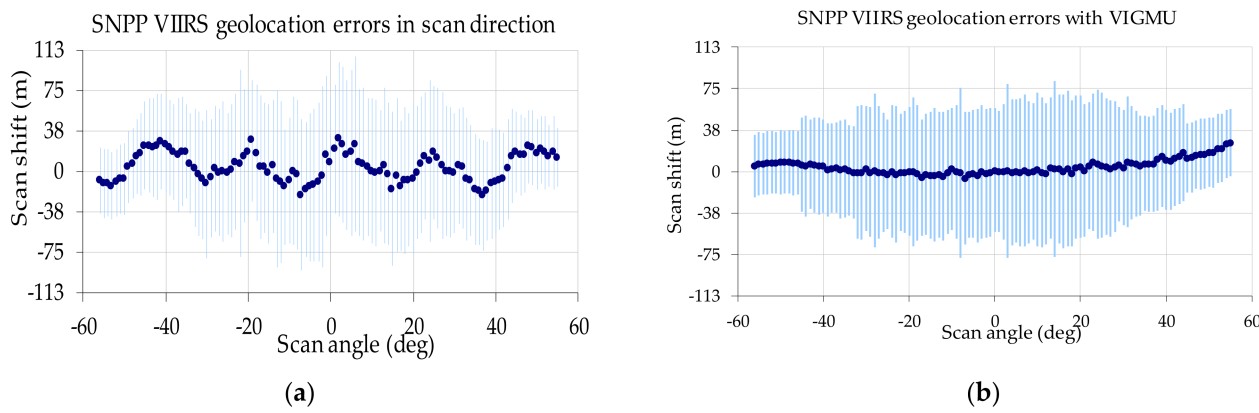

(**a**) (**b**)

**Figure 8.** (**a**) SNPP VIIRS geolocation shift in the scan direction in C1 before VIGMU implementation. (**b**) After. Dots represent the means in 1-degree bins while the error bars represent the standard deviations from the means within these bins from the geolocation error measurements.

### 4.3.3. Kalman Filter to Improve SNPP Attitude Due to Star Tracker Degradation

After the first few years of on-orbit operations, we started seeing large SNPP attitude excursions on a regular basis. We worked with the operations team and determined the issue was related to star tracker degradation. Cooling of star trackers' CCD arrays by lowering the thermal electric cooler (TEC) setpoint was implemented in September 2016 and proved to be effective as shown in Figure 9.

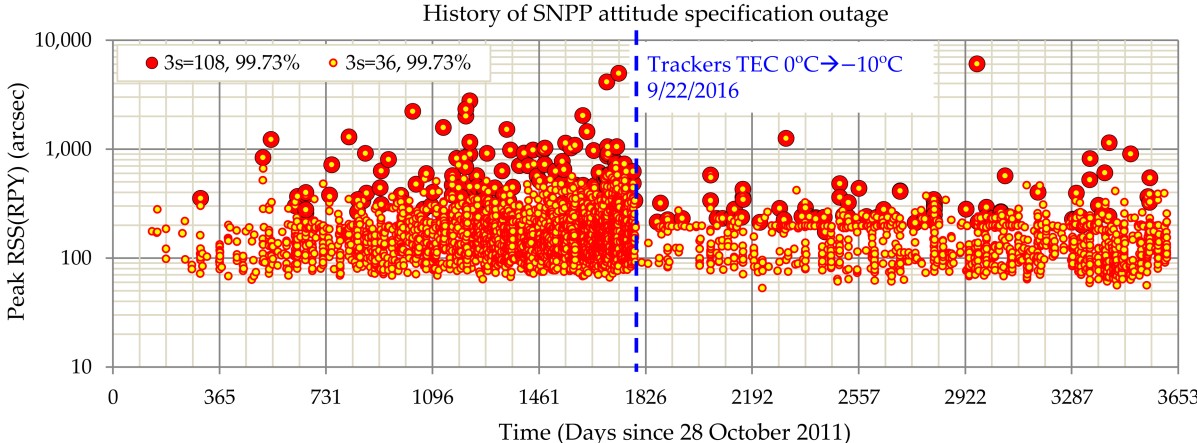

**Figure 9.** Trends of SNPP attitude disturbances. Large circles indicate control specification (108 arcseconds, 3-σ) outage per orbit. Small dots show knowledge specification (36 arcseconds, 3-σ, "3s" in the legend) outage. Star tracker cooling completed on 22 September 2016 (Day 1791 on the *x*-axis) improved SNPP attitude performance. Large attitude excursions due to known maneuvers are excluded. Incomplete orbit transitions one day to the next in the data archive are not included. Thus, the plot represents about 90% of all available data.

A case of attitude control specification outage is shown in the solid curves in Figure 10. They are reported from the spacecraft (SC) and downlinked to the ground to aid in geolocation processing. We used a ground-based Kalman filter [39] to estimate actual attitude using gyro data that were downlinked to the ground. Assuming that the Kalman filter-processed attitude data are true, then the differences between the attitude data from the Kalman filter (KF) and those from the spacecraft are the knowledge errors. We carried out a study [40] into this case by using the technique of land/water masking and by shifting the images in each scan to match the coastline segments in western Australia from the data 16 days earlier. The amount the image shifts is compatible with the knowledge error [29].

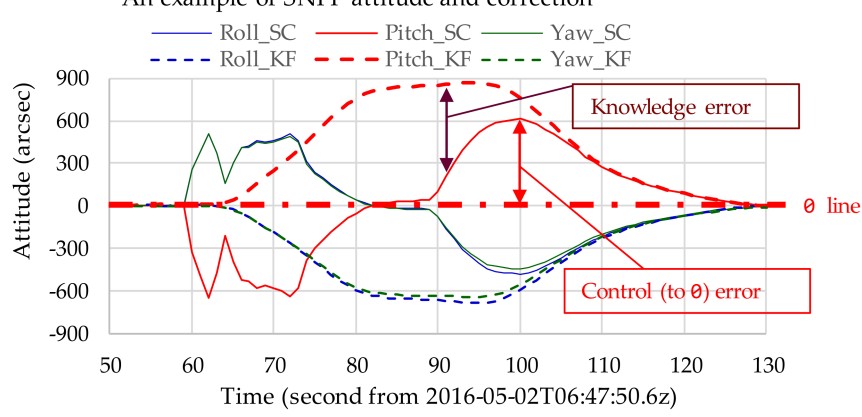

**Figure 10.** An example of SNPP attitude excursion and definitions of control and knowledge errors. The curves from Kalman filter processing are accurate and the differences are the knowledge errors.

In NASA VIIRS Land SIPS, we implemented the Kalman filter in the most recent SNPP VIIRS Collection-2 (C2) re-processing which started in November 2020, as well as in the continuing forward processing of C2 after April 2021. The Kalman filter has also been implemented in NASA Atmosphere SIPS. We are also seeking to implement the Kalman filter in other processing centers.

### 4.3.4. Correction for VIIRS Temporal Pointing Variations

Based on the geolocation trending experiences from MODIS sensors onboard the Terra and Aqua satellites [14,41], it was anticipated that there may be seasonal, annual, and/or other long-term trends of pointing variations mainly due to thermal effects and aging. VIIRS design tried to minimize these variations, which is mostly true for SNPP VIIRS, however, the NOAA-20 VIIRS sensor has exhibited regular annual pointing variation. These variations are corrected for by modifying the geolocation LUTs with the insertion of fields describing additional correction in the instrument-to-spacecraft mounting angles in roll, pitch, and yaw (RPY). The three fields are a count of valid points, a list of timestamps, and a corresponding list of RPY values. Code from the MODIS geolocation processing system was borrowed that reads the count variable, searches timestamps for the two bracketing times surrounding the current scan, and linearly interpolates the RPY values to the scan time. If the scan is beyond the edge of the array, it simply uses the most recent RPY values. With the RPY correction calculated for the scan, the code then converts to a rotation matrix and multiplies with the inst2sc matrix to obtain the adjusted inst2sc.

The correction for temporal pointing variations was first implemented in C2 reprocessing in NASA Land SIPS, starting in July 2019 for NOAA-20 VIIRS and November 2020 for SNPP VIIRS. The results are shown in Figures 5 and 6a for SNPP and NOAA-20, respectively. Further improvements are anticipated in future data re-processing.

## 5. Discussion and Future Work

Some VIIRS geolocation Cal/Val activities listed in Section 4 were not anticipated and thus not planned before launch. These activities include loss of synchronization from HAM to RTA, sector rotation for lunar calibration maneuvers, SNPP star tracker re-alignment, SNPP spacecraft control clock error, VIIRS instrument geometric model update, SNPP star tracker degradation, Kalman filter incorporation, and focal length measurement with on-orbit data. Anticipated activities to improve geolocation performance include terrain correction for DNB geolocation, correction for VIIRS temporal pointing variations, and trailing effects after active orbit management operations and delayed leap-second insertions.

For on-orbit geolocation calibration and validation, it was anticipated to update geolocation parameter LUTs every six months using the historical trends in forward processing after an intensive Cal/Val phase. Once we start re-processing, large geolocation biases in the past will be corrected if a method is developed and is viable. There are also cases where large geolocation biases cannot be corrected, such as those discussed in Section 4.1. NASA Land SIPS will then mask out the granules during these time periods.

Notice that the VIIRS sensor behaves differently in geolocation performance onboard SNPP and NOAA-20 satellites. The pointing variations are very regular in annual cycles on NOAA-20. However, they are irregular on SNPP. We will need to monitor geolocation variations closely for VIIRS sensors onboard JPSS-2 to JPSS-4 in the near future. These experiences should make the future VIIRS geolocation Cal/Val processes easier.

### 5.1. Future Work

Further improvements in geolocation performance have been planned to update the digital elevation model (DEM) from 1 km that is currently in use to finer and newer 500 m or 250 m resolution datasets [42]. In addition, annual land/water masks derived from MODIS observations are available now and will be implemented in future VIIRS data collections. Additionally, we are working on generating chip libraries using other Landsat-8 spectral bands that match additional VIIRS bands [26]. These multi-band chip libraries can be used to regularly monitor on-orbit band-to-band registration in the future.

Improvements in geolocation performance for future VIIRS sensors are expected based on the lessons learned from SNPP and NOAA-20 on-orbit operations, and/or JPSS-2 VIIRS sensor ground testing. These include higher sampling rate of attitude data from JPSS-2+ spacecrafts due to under-sampling from SNPP and NOAA-20, and scan rate increase for

VIIRS sensors on JPSS-3 and JPSS-4 due to scan-to-scan underlaps found from VIIRS sensors on JPSS-1 and JPSS-2. We may designate these two items as the results of risk reduction from the NPP project.

### 5.2. Higher Sampling Rate of Attitude Data from JPSS-2+ Spacecrafts

The sampling frequency of attitude data from the ADCS onboard the SNPP and NOAA-20 spacecrafts is at 1 Hz, i.e., one sample every one second of time. This sampling frequency is not able to capture oscillations close to or higher than its corresponding Nyquist frequency of 0.5 Hz. When that under-sampling happens, "beat frequency" could be observed. We observed such events in 2012 from SNPP attitude such as in Figure 11a. We suspected the rotation of VIIRS RTA at 0.56 Hz (rotation period of 1.8 s) might be one of the causes. Communications with the VIIRS vendor found that there is a slight RTA dynamic imbalance in the VIIRS build for SNPP. Subsequently, the RTA dynamic balancing was carried out more carefully for the VIIRS build for JPSS-1 (NOAA-20). However, the "beat frequency" from the NOAA-20 attitude oscillations still exists (Figure 11b), although the magnitude is smaller.

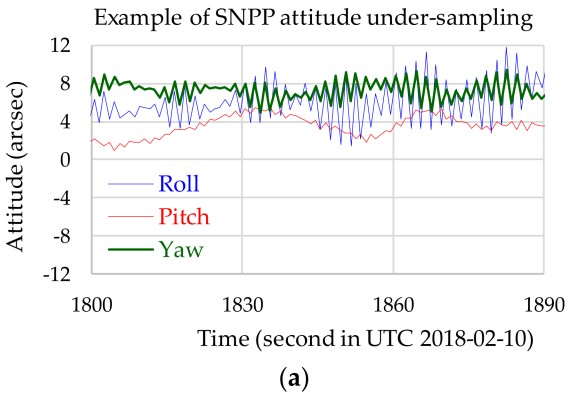
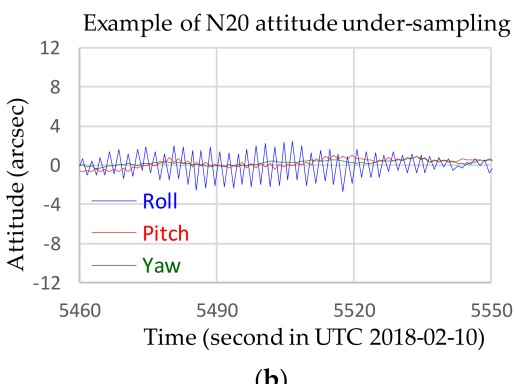

(**a**)                                                                                  (**b**)

**Figure 11.** Examples of attitude under-sampling in 1 Hz from (**a**) SNPP and (**b**) NOAA-20 ADCS at 1 Hz that result in "beat frequencies" and aliasing in roll and yaw directions for SNPP and in roll direction in NOAA-20.

One scan of VIIRS samples in the EV sector spans 112 degrees that takes 0.56 s. Geolocating these samples uses attitude data by linear interpolation of the adjacent two or three attitude samples. The under-sampling of rapidly varying attitude data causes a geolocation pointing error of about 5 arcsec in the roll direction for SNPP, which is about 20 m at nadir or about 130 m at the edges of scan in the scan direction for SNPP VIIRS. Considering other components onboard the JPSS spacecrafts that have mechanical motion close to 0.5 Hz, the sampling frequencies of attitude from the spacecrafts have been increased to 10 Hz for JPSS-2 to JPSS-4 that will capture likely attitude oscillations near and higher than 0.5 Hz. The higher sampling frequency of attitude data is expected to improve geolocation performance.

### 5.3. Faster Scan Rate for VIIRS Sensors on JPSS-3 and JPSS-4

The VIIRS instrument is required to have contiguous, non-underlapping scans at nadir. During JPSS-2 VIIRS BBR tests under ambient conditions, a difference was found between HAM side-0 and HAM side-1. A contribution to this difference is the misalignment of the mirror plane-of-symmetry to the HAM motor axis. This misalignment was deemed too large, which would result in non-compliance to the scan-to-scan overlap requirement. The underlap, or gap, occurs because the misalignment causes one scan to tilt backward in the ground track and the next scan to tilt forward, which would create a gap between the alternating scans. Consequently, a re-work was carried out to correct for the misalignment.

Curiosity led us to a study of the scan-to-scan overlap using on-orbit operational SNPP ephemerides. The study found that the original design missed the effect of Earth rotation

in the calculation of scan-to-scan overlap [43]. That caused underlap to occur in about 70 m at nadir for NOAA-20 VIIRS around 15°N, where the satellite altitude is the lowest and ground speed is the fastest [16]. Unfortunately, the scan-to-scan underlap also occurs to MODIS on both Terra and Aqua satellites [41].

The scan-to-scan underlaps are unexpected to occur in the nominal design of the VIIRS instrument. After the impact of Earth rotation is considered, effective focal length was adjusted for JPSS-3 and JPSS-4 VIIRS sensors. The new nominal effective focal length (EFL) is 1131.8 mm, a change of 0.81% from the original 1141 mm. The ground coverage in the scan direction will be extended (Table 5). The corresponding scan rate increases from 3.517 to 3.545 radians/s. The scan period becomes 1.7724 s from the original 1.7867 s, the change of which is required to meet band-to-band co-registration [8,10]. Note that the EFL and corresponding scan rate in actual built sensors may vary up to 0.5% [26] from the nominal values.

**Table 5.** Two sets of nominal EFLs and scan rates and EV coverages at altitude 828 km.

| Platform | EFL (mm) | Scan Rate (rad/s) | EV Scan Angle (deg) | EV Ground Distance (km) |
|---|---|---|---|---|
| SNPP, JPSS-1, JPSS-2 | 1141.0 | 3.517 | ±56.04 | ±1510 |
| JPSS-3, JPSS-4 | 1131.8 | 3.545 | ±56.50 | ±1550 |

This lesson learned from VIIRS design in the early JPSS Program has been applied to the VIIRS sensor builds in the last two JPSS satellites. We have also been seeking to inform other instrument designers of such a potential issue and the mitigation method in as many ways as possible.

## 6. Conclusions

The overall performance of SNPP and NOAA-20 on-orbit VIIRS geolocation is very good. After on-orbit calibration and improvement activities, sub-pixel geolocation accuracy is achieved. Nadir equivalent geolocation accuracy is generally within 75 m, or 20% nominal imagery band pixel size of 375 m, in either the along-scan or along-track direction for both SNPP and NOAA-20 VIIRS sensors in 16-day measurement statistics. The worst 16-day measured geolocation errors (radial, 3-$\sigma$) are 280 m and 267 m in the latest SNPP VIIRS C2 and NOAA-20 VIIRS C2.1, respectively, which are better than the required accuracy of 375 m (radial, 3-$\sigma$).

We have paid great attention to the detailed trending of the errors detected by the control point matching program. Having experiences from MODIS geolocation calibration and long-term trending helped us in identifying and mitigating issues, such as terrain correction for DNB geolocation, correction for temporal pointing variations, masking out data granules having large geolocation errors following DMUs and IAMs, and closely working with teams in orbit operations to minimize adverse impacts on geolocation performance.

We have also been on the look-out for possible improvements in future satellite performance. They include downlinking the spacecraft attitude in a higher sampling rate, and designing VIIRS-like instruments in a proper cross-track scan rate so as to avoid scan-to-scan underlaps.

**Author Contributions:** Conceptualization, G.L., R.E.W.; methodology, G.L., R.E.W., P.Z.; software, J.J.D., P.Z.; data curation, J.J.D., P.Z., B.T.; writing—original draft preparation, G.L.; writing—review and editing, P.Z., B.T., J.J.D.; visualization, G.L., P.Z. All authors have read and agreed to the published version of the manuscript.

**Funding:** This research received no external funding.

**Data Availability Statement:** NASA Land SIPS re-processed and forward-processed VIIRS Level-1 radiometry and geolocation datasets are publicly available via NASA Level-1 and Atmosphere Archive & Distribution System (LAADS) Distributed Active Archive Center (DAAC).

**Acknowledgments:** The authors acknowledge the cooperation and assistance from many colleagues such as those from Raytheon Company, the NASA VIIRS Land SIPS, NOAA Mission Operations Team (MOT) and JPSS Flight Project, and Fred Patt of Science Applications International Corporation (SAIC) on-site at NASA Goddard Space Flight Center.

**Conflicts of Interest:** The authors declare no conflict of interest.

## Appendix A  List of Abbreviations Used in This Paper

There are many abbreviations (acronyms) used throughout this paper. They are listed in Table A1 for easy reference.

**Table A1.** Abbreviations used in this paper.

| Acronym | Explanation | Acronym | Explanation |
|---------|-------------|---------|-------------|
| AOA | aft-optics assembly | LRM | lunar roll (calibration) maneuver |
| ADCS | attitude determination and control system | LTAN | local time at the ascending node |
| AVHRR | Advanced Very-High Resolution Radiometer | LUT | look-up table |
| Cal/Val | calibration and validation | M | moderate-resolution band in VIIRS |
| CCD | change-coupled device | MODIS | Moderate-resolution Imaging Spectroradiometer |
| CCV | correlation coefficient value | MOT | Mission Operations Team |
| CDR | climate data record | N20 | NOAA-20 satellite (JPSS-1 before launch) |
| CPM | control point matching | NG | Northrop Grumman |
| DEM | digital elevation model | NPOESS | National Polar-Orbiting Operational Environmental Satellite System |
| DNB | Day–night band (in VIIRS instrument) | NPP | NPOESS Preparatory Project |
| DMSP | Defense Meteorological Satellite Program | NRT | near-real time |
| DMU | drag make-up maneuver | NWP | numerical weather prediction |
| ECI | Earth center inertial (reference frame) | OLS | Operational Linescan System |
| EDR | environmental data record | PEATE | Land Product Evaluation and Testing Element |
| EFL | effective focal length | PGE | product generation executable |
| EOC | early orbit checkout | POES | Polar-Orbiting Environmental Satellite |
| EOS | Earth Observing System | RMSE | root mean square error |
| ESDR | Earth science data record | RPY | roll, pitch, and yaw |
| EV | Earth View | RSS | root sum squared |
| FPA | focal plane assembly | RTA | rotating telescope assembly |
| GCP | ground control point | SC | spacecraft |
| GRAVITE | Government Resources for Algorithm Verification Independent Test and Evaluation | SCE | scan control electronics |
| GPS | Global Positioning System | SDR | sensor data record |
| HAM | half-angle mirror | SeaWiFS | Sea-viewing Wide Field-of-view Sensor |
| HSI | horizontal sampling interval | SIPS | Science Investigator-led Processing System |
| I | imagery resolution band in VIIRS | SNPP | Suomi National Polar-orbiting Partnership |
| IAM | inclination adjust maneuver | STAR | Satellite Applications and Research |
| IDPS | Interface Data Processing Segment | SV | Space View |
| J1 | JPSS-1 | UTC | Coordinated Universal Time |
| JPSS | Join Polar Satellite System | VIGMU | VIIRS instrument geometric model update |
| KF | Kalman filter | VIIRS | Visible Infrared Imaging Radiometer Suite |

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
