# Peer review of "Ten Years of VIIRS On-Orbit Geolocation Calibration and Performance"

_remotesensing, doi:10.3390/rs14174212_

Round 1
Reviewer 1 Report
This is my review on "Ten Years of VIIRS On-Orbit Geolocation Calibration and
Performance" submitted to Special Issue Frontiers in Remote Sensing Techniques and Applications Using Visible Infrared Imaging Radiometer Suites
In this manuscript, the authors basically report the on-orbit geolocation issues of VIIRS sensors (about 0.4-12 mu) on a time span of 10-years for two
different missions (SNPP, NOA-20).
The resulting data generated by these instruments are essential for climate
prediction and generation of products with global validity.
Moreover, the technical details associated to on-orbit performances are not only interesting to report, but necessaries for an adequate validation and
confiability on the final data and products.
Hence, the subject is very important and timely for the Special Issue at hand.
The subject is very interdisciplinary, involving astrodynamics, engineering,
solid-state physics, etc. Hence, different audiences
could be more/less interested in different issues. Taking this into account,
the manuscript is well-balanced, well written and follows all the standards in
this field. For example, is particularly interesting the discussion about the
leap-second insertion (due to the changes in TUC -coordinate time-), it is
an issue of hot debate in astrometry.
The manuscript can be accepted in its present form.
But I'm curious about a couple of things which I would like to know, or even could be interesting to explain in more detail for interested readers.
E.g.,
> Page 4., line. 170, the authors refer to a "change in the reference frame".
It is not clear to me about which frame are they referring
(orbital, terrestrial, instrumental, etc...) please, can you explain this?
Another comment: the title is very similar to this already published article: https://www.mdpi.com/2072-4292/13/20/4179/htm
which can cause some bibliographic drawbacks.
Reviewer 2 Report
This is a thorough and clearly written manuscript describing the geolocation calibration efforts of the past 10 years across two satellite platforms. This manuscript is especially timely given that JPSS-2 will be launched later this year.
I have little to add in terms of scientific comment and merit. The many figures showing errors over time, scan angle, and before/after examples are very useful to illustrate the lessons learned over the decade.
In terms of readability, it is well written and organized, but I have a few minor suggestions. While the topic of the paper is geolocation calibration, I do not see a clear definition of what the authors mean by geolocation. As an example, the VIIRS SDR Users Guide provides this:
Knowledge of the Earth location (geolocation) latitude and longitude is essential to put the VIIRS data in a geographic context and with other spatially referenced data sets, including other VIIRS data, and to provide a uniform, worldwide spatial reference system for all data products.
Given that this manuscript is partially a historical/review article, I think the authors should similarly add a sentence or two describing the problem in the introduction.
Line 94 - I think the authors should highlight that this acronym appendix is available. This paper is very dense in acronyms, and I referenced the acronym list many times. When I am reading paper, I often skip over the sentences that describe the paper outline. The appendix was mentioned at the end, so I missed it on my first read through. Perhaps the authors should note the useful appendix just after line 88 and before the paper outline on line 89.
Line 98 - while I understood what a chip was based on the context of the sentence, I think the authors should define what is meant by chip.
Line 191 - roll, pitch and yaw parameters are shown here, and again in Figure 4 and line 431. However, it's not until line 541 probably that these are described as the instrument-to-spacecraft mounting angles. For clarity, I recommend defining the terms roll, pitch and yaw as the instrument-to-spacecraft mounting angles before line 191 and the rest of the text can remain as-is. Another possible location could be the first mention of the ADCS, on line 154.
My edits are very minor, but my hope is to make this topic as accessible as possible. I recommend publication of this paper almost as-is.
